# The Zinc Finger Protein Zfp2 Regulates Cell–Cell Fusion and Virulence in *Cryptococcus neoformans*

**DOI:** 10.3390/jof11120868

**Published:** 2025-12-07

**Authors:** Cheng-Li Fan, Lin Li, Ji-Chong Shi, Tong-Bao Liu

**Affiliations:** 1College of Animal Science and Technology, Southwest University, Chongqing 400715, China; victoriafan@swu.edu.cn; 2Wanzhou District Center for Disease Control and Prevention, Chongqing 404199, China; liin12181204@163.com; 3Medical Research Institute, Southwest University, Chongqing 400715, China; sjc112023@email.swu.edu.cn

**Keywords:** *Cryptococcus neoformans*, zinc finger protein, Zfp2, cell fusion, virulence

## Abstract

*Cryptococcus neoformans* is a fungal pathogen commonly found in the environment. It mainly infects immunocompromised individuals, causing cryptococcal pneumonia and meningitis, which result in hundreds of thousands of deaths each year. Zinc finger proteins, with zinc finger domains, are common across organisms and serve many biological functions. In this study, we identified and characterized Zfp2, a C_3_HC_4_-type zinc finger protein, which regulates cell fusion and virulence in *C. neoformans*. Stress tests showed that the *zfp2*Δ mutant is hypersensitive to SDS, Congo red, NaCl, KCl, caspofungin, and fluconazole, suggesting that Zfp2 helps maintain cell membrane or wall integrity in *C. neoformans*. Notably, deleting *ZFP2* reduced capsule size, while its overexpression led to capsule enlargement. The *zfp2*Δ mutants also demonstrated a growth defect at 37 °C. Cell fusion assay showed that Zfp2 is essential for cell fusion during sexual reproduction, as *zfp2*Δ mutants could not fuse during bilateral mating. To understand why the *zfp2*Δ mutants failed to fuse, we examined key genes in the pheromone response pathway and found that Zfp2 may affect cell fusion by regulating this pathway. Finally, a virulence test in mice showed that both *ZFP2* deletion and overexpression significantly reduced *C. neoformans’* virulence. Overall, our research suggests that the zinc finger protein Zfp2 is vital for cell fusion and virulence in *C. neoformans*.

## 1. Introduction

*Cryptococcus neoformans* is a pathogenic yeast widely distributed in the environment [1]. It mainly infects immunocompromised individuals, causing cryptococcal pneumonia and meningitis, which kill hundreds of thousands yearly [2,3]. In recent years, as the number of immunocompromised people rise, the rates of illness and death from cryptococcal infection have also grown [4,5]. As a human pathogenic fungus, *C. neoformans* has key virulence factors, including capsule formation [6,7], melanin production [8], laccase synthesis [8,9], and growth at 37 °C [10]. Besides its medical significance, *C. neoformans* has become a model organism for studying fungal genetics, being a rapidly reproducing haploid yeast with a defined sexual cycle [11]. 

*C. neoformans* is a basidiomycete with two mating types, *MAT*α and *MAT***a**, determined by a single mating-type locus (*MAT*) [12,13]. It usually propagates by budding but can switch to filamentous growth via mating or monokaryotic fruiting, triggered by environmental factors [14,15]. Mating begins when pheromones from one mating type cell are recognized by specific pheromone receptors on another mating type cell, activating the mitogen-activated protein (MAP) kinase pathways and promoting conjugation tube formation and size increase for cell fusion [16]. Following the cytoplasmic fusion, dikaryotic filaments form, leading to a basidium where meiosis occurs and produces four basidiospore chains. Monokaryotic fruiting, involving the same mating-type cells without clamp cells, resembles mating but occurs mainly in *C. deneoformans* (serotype D) [17,18].

Zinc finger proteins, first identified in *Xenopus oocytes*, constitute a major protein class characterized by zinc finger domains. These proteins mainly serve as transcription factors [19] and are present across eukaryotes, including plants like rice, Arabidopsis, and soybean, and fungi such as the rice blast fungus and *Saccharomyces cerevisiae* [20,21,22,23]. They play a key role in fungal conidial formation. For example, the *COS1* gene in *Magnaporthe grisea* is essential for conidial production, as deletion mutants fail to produce conidia, whereas complemented strains develop normally [24]. The zinc-finger transcription factor, Flb4p, is also involved in sporulation [25]. Moreover, the lack of the calcineurin transcription factor Crz1 markedly reduces conidia formation in *Beauveria bassiana* and delays germination [26].

Zinc finger proteins are crucial in fungal growth, development, and stress responses. In *S. cerevisiae*, the zinc finger proteins Msn2p and Msn4p activate the STRE-dependent promoters, with deletion of *MSN2* and *MSN4* increasing stress susceptibility [27,28,29]. Additionally, the zinc finger protein CrzA mediates osmotic stress responses in *Aspergillus nidulans*, especially to Mn^2+^ and Ca^2+^ [30], while the zinc finger protein AslA regulates potassium (K^+^) stress responses [31]. In *C. deneoformans*, the zinc finger protein Znf2, a transcription factor with four C_2_H_2_ zinc finger domains, regulates yeast-to-hyphal transition and hyphal morphogenesis post-cell fusion during mating, while Znf3, containing three C_2_H_2_ zinc finger domains, is essential for both unisexual and bisexual reproduction [32,33]. Our previous research shows Zfp1, a zinc finger protein, controls sexual reproduction and virulence in *C. neoformans* [34]. Overall, zinc finger proteins constitute an important class of regulatory proteins that significantly influence fungal growth and development and play vital roles in their resilience against environmental stresses. However, due to the large number and varied structures of zinc finger proteins, their roles in fungal morphogenesis and pathogenicity still need further investigation.

Building on our prior iTRAQ-based proteomic analysis of autophagy pathway proteins revealed differential gene expression in *atg8*Δ and H99 strains (data not published). We prioritized CNAG_06324, a zinc finger domain-encoding gene designated Zfp2. In this study, we demonstrate that the C_3_HC_4_-type zinc finger protein Zfp2 regulates key traits in *C. neoformans*; *zfp2*Δ mutants exhibit heightened sensitivity to stressors, implicating its role in cell membrane/wall integrity; it modulates capsule size by reducing it in knockouts and enlarging it upon overexpression, demonstrating its dose-dependent role in this virulence-associated trait; and it is essential for sexual reproduction via cell fusion, as deletion blocks this process entirely. Mechanistic insights suggest that Zfp2 regulates fusion through the pheromone response pathway, while virulence assays confirm that *zfp2*Δ mutants display attenuated pathogenicity in mice. These findings establish Zfp2 as a critical determinant of cell fusion and virulence.

## 2. Materials and Methods

### 2.1. Strains and Growth Conditions

*C. neoformans* strains and plasmids used in this study are listed in Appendix A [34,35,36,37,38]. They are usually grown on YPD medium (1% yeast extract, 2% peptone, and 2% dextrose) at 30 °C. Melanin production and capsule formation are stimulated by L-DOPA and diluted Sabouraud medium, respectively, as previously described [39,40]. Mating and sporulation are induced on MS medium and V8 medium, prepared according to established protocols, and maintained at 25 °C in the dark [39]. Additionally, YPD medium supplemented with various chemicals was used to assess growth under stress, following established protocols [39].

### 2.2. RT-qPCR

To measure gene expression in *C. neoformans*, we used RT-qPCR to validate the mRNA transcription levels. Cultures or mating mixtures of *C. neoformans* strains were harvested and lyophilized. Total RNA was extracted using the SteadyPure Universal RNA Extraction Kit (AGBio, Changsha, China) following the manufacturer’s instructions. cDNA was synthesized using the Evo M-MLV RT Mix Kit (AGBio, Changsha, China) with gDNA Clean for qPCR. Amplification was performed on the LineGene 9600 Plus qPCR system (Bioer, Hangzhou, China) with gene-specific primers (see primer information in Appendix A) and SYBR Green Premix (AGBio, Changsha, China). Gene expression was normalized to the endogenous control gene *GAPDH*, and relative expression was determined using the 2^−ΔΔCT^ method [39]. Statistical significance was evaluated with one-way ANOVA followed by Tukey’s multiple comparison test (GraphPad Software Inc., San Diego, CA, USA).

### 2.3. Generation of GFP-Tagged Strains

To determine Zfp2 protein localization in *C. neoformans*, the *ZFP2* gene’s coding sequence (705 bp) was amplified from wild-type H99 DNA using primers TL1124 and TL1125. The fragment was then inserted into the pCN19 vector [38] to create GFP-Zfp2 fusion construct, named pTBL195. After *Sca*I linearization, the plasmid was concentrated to 2 μg/μL and introduced into the *zfp2*Δ mutant strain via biolistic particle bombardment. Transformants were selected on YPD agar plates containing 100 μg/mL nourseothricin sulfate, and GFP-positive clones were verified with a Zeiss Axio Observer 3 fluorescence microscope equipped with a 100× objective lens (Axio Observer 3; Zeiss, Oberkochen, Germany). The GFP-Zfp2 strain was designated TBL344.

### 2.4. FM4-64 and DAPI Staining

To identify Zfp2 localization in *C. neoformans*, we performed FM4-64 and DAPI staining. FM4-64 (Sigma, Saint Louis, MO, USA) was diluted with anhydrous dimethyl sulfoxide (DMSO) to 20 ng/μL, stored in the dark at −20 °C. Cells were harvested and washed twice with 1× PBS buffer, centrifuged, and resuspended in 1× PBS to the desired cell density. Then, 10 μL of cells was mixed with 100 μL of FM4-64, incubated on ice for 50 s to label membranes. After incubation, the sample was centrifuged at 8000 rpm for 2 min to pellet the cells. Then, 95 μL of supernatant was removed, leaving a small volume to resuspend the cell pellet. Finally, 1 μL of the stained cells was placed onto a microscope slide, covered with a coverslip, and observed under an Axio fluorescence microscope (Axio Observer 3; Zeiss, Oberkochen, Germany) with FM4-64 filters.

To visualize cell nuclei, DAPI staining was performed following a modified protocol as previously described [41]. Briefly, 500 μL of the overnight cultures were collected and rinsed twice with 1 mL of sterile 1× PBS (pH 7.4). The cells were fixed with 200 μL of 9.3% formaldehyde at 25 °C for 10 min, then rinsed twice with 1 mL of 1× PBS buffer. For permeabilization, the pellet was resuspended in 300 μL of 1× PBS, then mixed with an equal volume of 1× PBS with 1% Triton X-100, and incubated for 10 min at room temperature. After washing twice with 1 mL of 1× PBS, cells were resuspended in 300 μL of PBS, and an equal volume of a 10 μg/mL DAPI (Sigma, Saint Louis, MO, USA) was added. The cells were incubated in the dark at room temperature, shaking at 60 rpm for 15 min to facilitate DAPI binding to nuclear DNA. Finally, the stained cells were rinsed twice with 1 mL of PBST buffer, resuspended, and observed under an Axio fluorescence microscope with a 100× objective lens.

To examine the subcellular distribution of the Zfp2 protein under different stress conditions, the GFP-Zfp2 fluorescent yeast strain was grown in YPD medium with specific stressors. These included agents affecting cell integrity (0.025% SDS and 0.5% Congo red), osmotic stress (1.5 M NaCl and 1.5 M sorbitol), and oxidative stress (2.5 mM H_2_O_2_). The cultures were grown at 30 °C with agitation at 200 rpm for 24 h. Afterwards, GFP-Zfp2 localization was analyzed using an Axio fluorescence microscope equipped with a GFP filter set and a 100× oil-immersion objective lens. Simultaneously, differential interference contrast (DIC) imaging was performed to evaluate cell morphology.

### 2.5. Generation of ZFP2 Deletion, Complementation, and Overexpression Strains

The *ZFP2* gene was knocked out in *C. neoformans* strains H99 and KN99**a** using a split-marker strategy. Upstream (TL837/TL838) and downstream (TL839/TL840) *ZFP2* fragments were PCR-amplified from the H99 DNA (all primers used in this study are listed in Appendix A). The *NEO* resistance gene was amplified from plasmid pJAF1 [42] using primers TL17/TL18. Next, fusion PCR was performed to generate the 5′ and 3′ recombination fragments. The 5′ fusion fragment (upstream *ZFP2* sequence fused to *NEO*) was amplified with primers TL837/TL20, using the upstream *ZFP2* fragment and *NEO* as templates. Similarly, the 3′ fusion fragment (*NEO* fused to the downstream *ZFP2* sequence) was amplified with primers TL19/TL840. Equal amounts of the 5′ and 3′ fusion fragments were mixed, coated onto gold microcarriers, and introduced into H99 and KN99**a** strains via biolistic transformation. Transformants were selected on YPD plates containing 200 mg/L G418. Positive transformants were confirmed by PCR using primers TL843/TL59 (to detect the integrated *NEO* gene), while negative controls were verified with primers TL841/TL842 (specific to the wild-type *ZFP2* locus). Finally, Southern blot analysis was performed to confirm the proper deletion of the *ZFP2* gene.

To create the *zfp2*Δ::*ZFP2* complementation strain, a genomic fragment with native *ZFP2* promoter, open reading frame (ORF), and terminator was amplified from H99 DNA using primers TL1098/TL1099. This fragment was cloned into the plasmid pTBL1 [34], resulting in the complementation vector pTBL183. The plasmid pTBL183 was linearized with *Sac*II and introduced into *zfp2*Δ mutants in H99 and KN99**a** backgrounds. Confirmations were performed via PCR, sequencing and phenotypic restoration to verify successful complementation.

For *ZFP2* overexpression, the *ZFP2* ORF was amplified from H99 genomic DNA using primers TL1100/TL1101 and cloned into plasmid pTBL153 which contains a constitutive Actin promoter to create the overexpression vector pTBL211. The pTBL211 plasmid was cut with *Xho*I and introduced into the H99 and KN99**a** strains. To confirm overexpression, RT-qPCR was performed on the selected transformants to measure *ZFP2* mRNA levels relative to a reference *GAPDH* gene.

### 2.6. Virulence Factor Production Assay and Growth Under Stress Conditions

To evaluate melanin synthesis in *C. neoformans* strains, serial dilutions of yeast cells were prepared from overnight cultures. A 5 μL volume of each dilution was spotted onto L-DOPA (L-3,4-dihydroxyphenylalanine) agar medium, which induces melanin production through laccase activity. The plates were incubated at 30 °C or 37 °C for 2–3 days. Melanin production was visually assessed by observing the development of dark brown to black pigmentation around the colonies. Representative images were captured, and pigment intensity was qualitatively compared across strains.

To analyze capsule polysaccharide synthesis, *C. neoformans* strains were cultured overnight in YPD broth at 30 °C. Cells were harvested by centrifugation, washed twice with 1× PBS, and resuspended in diluted Sabouraud dextrose broth to induce capsule formation [40]. The cultures were incubated overnight at 37 °C with shaking at 150 rpm. Capsule size was measured through microscopic analysis. Cells were combined with India ink and examined under a light microscope at 400× magnification. For each strain, capsule thickness was measured in at least 100 individual cells using ImageJ (v1.52). The average capsule diameter ± standard deviation was calculated to compare capsule production across strains. Statistical significance was determined using an ANOVA test.

To study Zfp2’s role in *C. neoformans* stress resistance, we examined the growth of various cryptococcal strains on YPD plates with stressors. Overnight cultures were washed three times with PBS, diluted to an OD_600_ of 2.0, and serial ten-fold dilutions were made in ddH_2_O. A 5 μL volume of each dilution was spotted onto plates, then incubated at 30 °C for 2–3 days. Growth was documented by photographing. 

To evaluate Zfp2’s effect on growth at 37 °C, we started with an inoculum at an OD_600_ of 0.1. Then, 100 μL of this culture was incubated at 37 °C for 12 h, with OD_600_ readings taken every 2 h. The optical density was measured using a Denovix DS-11 Spectrophotometer (Denovix, Wilmington, DE, USA).

### 2.7. Fungal Mating Assays

To evaluate mating compatibility and basidiospore production, we conducted cross-mating assays using *C. neoformans* strains of opposite mating types. Overnight cultures of *MAT*α and *MAT***a** strains were grown in YPD broth at 30 °C. Cells were collected by centrifugation, washed twice with sterile distilled water, and resuspended to a final concentration of 1 × 10^8^ cells/mL. Equal volumes (50 μL each) of *MAT*α and *MAT***a** cell suspensions were mixed thoroughly and spotted onto mating-inducing MS or V8 media. Plates were incubated in the dark at 25 °C for 10–14 days to promote mating. After incubation, mating filaments (hyphae) and basidiospore chains were visualized using an Olympus CX41 light microscope (Olympus, Tokyo, Japan) at 40×, 200×, and 400× magnifications. Digital images were captured for documentation.

### 2.8. Cell–Cell Fusion Assay

To evaluate mating efficiency and cell fusion capacity, a quantitative fusion assay was performed as previously described with modifications [43]. Briefly, strains of opposite mating types were grown overnight in YPD broth at 30 °C, harvested, washed, and resuspended in YPD to 2 × 10^6^ cells/mL. Equal volumes of *MAT***a** and *MAT*α suspensions were mixed. A 5 μL aliquot was spotted onto V8 juice agar medium (pH 5.0) to induce mating, then incubated at 25 °C in the dark for 24 h. Cells were scraped from the agar surface, resuspended in 1 mL of sterile water, and plated onto selective medium containing nourseothricin (NAT, 100 μg/mL, Sango, Shanghai, China) and G418 (200 μg/mL, Sango, Shanghai, China) to select for fusion products. Colonies were counted after 3–5 days of incubation at 30 °C, with wild-type fusion (*MAT*α *NAT* × *MAT***a** *NEO*) efficiency set as 100%, and mutant fusion efficiency calculated as: relative fusion efficiency (%) = (mutant colonies /wild-type colonies) × 100. Statistical analysis (ANOVA) was performed to compare fusion efficiencies between strains. All experiments were conducted at least three times with biological replicates.

### 2.9. Virulence Studies

To assess Zfp2’s role in fungal virulence, we infected mice using a well-established cryptococcosis model. *C. neoformans* strains were cultured overnight in YPD broth at 30 °C, then washed twice with sterile 1× PBS, and resuspended in PBS at 2 × 10^6^ cells/mL. Female C57BL/6 mice (6–8 weeks old, 18–22 g; Hunan SJA Laboratory Animal, Changsha, China) were randomly divided into experimental groups (*n* = 10 per group). Mice were anesthetized with isoflurane and intranasally inoculated with 50 μL of the yeast suspension (1 × 10^5^ CFU per mouse). The mice were monitored daily for signs of infection. Mice showing severe distress, moribundity, or more than 20% weight loss were euthanized via CO_2_ inhalation, following institutional animal care guidelines. Surviving mice were also euthanized at the endpoint. To further explore Zfp2’s role in fungal pathogenicity, we assessed virulence of Zfp2-related strains using the *Galleria mellonella* (wax moth larva; Keyunbio, Henan, China) model, following previously published protocols [44]. Survival data were plotted as Kaplan–Meier curves and statistically analyzed with the log-rank (Mantel–Cox) test in PRISM 9.0. A *p*-value less than 0.05 was regarded as statistically significant.

### 2.10. Fungal Burdens and Histopathological Examination in Infected Organs

At the experimental endpoint or when mice reached humane endpoints (as described in Section 2.10), mice were euthanized using CO_2_ inhalation. The brain, lungs, and spleen were aseptically dissected, weighed, and transferred to sterile 10 mL microcentrifuge tubes containing 1 mL of PBS. Organs were homogenized with a tissue grinder until a uniform suspension was formed. Serial dilutions of the tissue homogenates were prepared in PBS, and 100 μL of each dilution was spread onto YPD agar plates supplemented with ampicillin (100 μg/mL) and chloramphenicol (50 μg/mL) to inhibit bacterial growth. Plates were incubated at 30 °C for 48 h, and fungal colonies (CFU) were counted.

For histopathological analysis, portions of the brain, lungs, and spleen were fixed in 10% neutral-buffered formalin for 24–48 h. They were embedded in paraffin wax, sectioned at 5 μm, and stained with hematoxylin and eosin (H&E) according to standard protocols. Tissue sections were examined under an Olympus BX53 microscope (Olympus, Tokyo, Japan) at 200× and 400× magnification. Pathological features, including inflammatory cell infiltration, tissue necrosis, and the presence of fungal yeast cells, were recorded. 

Fungal burden data (CFU/g) were log-transformed to normalize distributions and compared between groups using one-way ANOVA with Tukey’s test in PRISM 9.0 (GraphPad Software). A *p*-value < 0.05 was considered statistically significant.

### 2.11. Serum Treatment and Cryptococcus–Macrophage Interaction Assay

To assess the viability of *C. neoformans* in mammalian serum, we conducted a serum survival assay. Overnight cultures of *C. neoformans* strains were collected by centrifugation, washed twice with sterile distilled water, and then resuspended in mouse serum to reach a final concentration of 1 × 10^6^ cells/mL. The cell-serum mixture (450 μL per well) was incubated at 37 °C in a humidified environment. At designated time points (0, 1, 2, 3, and 4 h), 50 μL samples were taken, serially diluted in PBS, and plated (100 μL per dilution) onto YPD agar. Plates were incubated at 30 °C for 48 h, and colony-forming units (CFU) were counted to assess cell viability. Survival rates were calculated as the percentage of viable cells relative to the initial inoculum.

To evaluate intracellular proliferation and phagocytic survival, we co-cultured *C. neoformans* with J774 murine macrophages. Macrophages were seeded in 24-well plates at 2 × 10^5^ cells per well in DMEM supplemented with 10% FCS and incubated overnight at 37 °C with 5% CO_2_ for adherence. *C. neoformans* strains were prepared as previously described and opsonized with 10% mouse serum for 30 min at 37 °C to enhance phagocytosis. Opsonized yeast cells were added to macrophages at a multiplicity of infection (MOI) of 5:1 (yeast/macrophage) and co-incubated for 2 h at 37 °C with 5% CO_2_ to facilitate uptake. Non-adherent extracellular yeast cells were rinsed away with warm PBS. Fresh DMEM (500 μL per well) was then added, and cultures were incubated for 0, 2, or 24 h. At each time point, macrophages were lysed by adding 200 μL of sterile distilled water for 5 min to release intracellular yeast. The lysate was serially diluted in PBS, and 100 μL aliquots were plated onto YPD agar. Plates were incubated at 30 °C for 48 h to count CFU.

Data were analyzed using PRISM 9.0 (GraphPad Software). Serum survival rates and Intracellular Proliferation Index (CFU at 24 h/CFU at 2 h) values were compared between strains using one-way ANOVA with Tukey’s test. A *p*-value < 0.05 was considered statistically significant.

## 3. Results

### 3.1. Identification of Zinc Finger Protein Zfp2

Our earlier research used an iTRAQ method to identify proteins involved in the autophagy pathway, revealing differential gene expression in the *atg8*Δ and H99 strains (Table 1). Among these genes, we focused on CNAG_06324 (Figure 1A), which encodes a zinc finger protein. Database search and structure prediction showed that it consists of 135 amino acids, with a C_3_HC_4_-type zinc finger near its N-terminus, named Zfp2 (Figure 1B). Compared to the previously studied Zfp1 [34], which spans 690 amino acids and contains three C_2_H_2_ zinc finger domains (Figure 1B), Zfp2 exhibits distinct sequence and structural features, highlighting its unique functional profile in the pathogen.

Given the limited research on C_3_HC_4_-type zinc finger proteins in fungi and their significance, we aimed to study the function of Zfp2 in *C. neoformans*. Our goal is to enhance our understanding of zinc-related roles in *C. neoformans* by investigating the C_3_HC_4_-type zinc finger protein Zfp2, which is important for basic research on this protein family. We first identified the subcellular location of Zfp2 in *C. neoformans*. We cloned the *ZFP2* gene into the pCN19 plasmid with a GFP tag, creating the GFP-Zfp2 fusion vector (pTBL195). This vector was then linearized and introduced into the *zfp2*Δ mutant strain. Localization analysis showed that Zfp2 is evenly distributed throughout the cytoplasm, as indicated by the fusion protein (Figure 1C). To further confirm this cytoplasmic localization of GFP-Zfp2, we used specific nuclear staining with DAPI and a membrane dye, FM4-64, to observe the GFP-Zfp2 fluorescent strain (Figure 1C). We also examined the localization of GFP-Zfp2 under various stress conditions, including high temperature (37 °C), osmotic stress (1.5 M NaCl, 1.5 M KCl, 1.5 M sorbitol), cell wall stress (0.5% Congo red), cell membrane stress (0.025% SDS), nitrosative stress (1 mM NaNO_2_ at pH 4.0), and oxidative stress (2.5 mM H_2_O_2_). In all these conditions, GFP-Zfp2 consistently localized to the cytoplasm (Figure 1D). Our results demonstrate that GFP-Zfp2 reliably localizes throughout the cytoplasm.

### 3.2. Zfp2 Is Involved in Capsule Formation and Cell Wall/Membrane Integrity in C. neoformans

*C. neoformans* has main virulence factors: capsule formation, melanin production, and the ability to grow at mammalian body temperature. To determine whether Zfp2 is involved in producing these virulence factors in *C. neoformans*, we created *ZFP2* gene knockout strains (*zfp2*Δ::*NEO*, TBL237 and TBL238, Appendix A), *ZFP2* complemented strains (*zfp2*Δ::*ZFP2*, TBL305 and TBL359, Appendix A), and *ZFP2* overexpression strains (*ZFP2*^OE^, TBL386 and TBL365, actin promoter, Figure 2A) in both *MAT*⍺ and *MAT***a** mating types. We first examined the in vitro formation of capsules by these strains. Compared with the wild-type strain H99, deletion of the *ZFP2* gene results in *C. neoformans* forming smaller capsules (Figure 2B,C). Conversely, overexpression of *ZFP2* results in larger capsules (Figure 2B,C).

Furthermore, we tested the growth of *ZFP2*-related strains at 30 °C and 37 °C and found that the *zfp2*Δ mutant strains showed a significant growth defect at both temperatures, with a more notable defect at 37 °C (Figure 2D). Additionally, the growth of *zfp2*Δ mutant strains was evaluated in liquid YPD medium at 37 °C (Figure 2E). The results showed that the *zfp2*Δ mutant’s growth rate was notably slower than that of other strains at 37 °C. 

Besides examining the influence of Zfp2 on the main virulence factors of *C. neoformans*, we also explored its role in the organism’s response to specific stress conditions in vitro. Notably, deleting the *ZFP2* gene made *C. neoformans* much more sensitive to various stressors, including 0.025% SDS, 0.5% Congo red, 1.5 M NaCl, 1.5 M KCl, and 1.5 M sorbitol (Figure 2D). *zfp2*Δ mutants also exhibited growth defects when exposed to cell wall antifungal agents caspofungin and membrane-active fluconazole (Figure 2D). These findings suggest that removing the *ZFP2* gene significantly weakens the structural integrity of both the cell membrane and cell wall of *C. neoformans*. 

### 3.3. Zfp2 Plays a Role in the Cell–Cell Fusion Process During the Sexual Reproduction of C. neoformans

*C. neoformans* has two distinct mating types, called *MAT*α and *MAT***a**, which can undergo sexual reproduction under certain nutritional conditions, leading to the formation of mating hyphae and eventually basidiospores. To explore the function of Zfp2 in the sexual reproduction of *C. neoformans*, we generated the *zfp2*Δ mutant and *ZFP2*^OE^ strain based on the H99 and KN99**a** backgrounds. These strains underwent both unilateral and bilateral mating assays. Notably, both types of mating in the *zfp2*Δ mutants failed to occur as effectively as in wild-type strains (Figure 3A). Interestingly, although one-sided mating involving the *ZFP2*^OE^ strain (TBL386 × KN99**a**) was capable of successful sexual reproduction, all other mating conditions for the *ZFP2*^OE^ strains did not produce mating hyphae under standard conditions (Figure 3A). These observations suggest that the presence and expression level of Zfp2 are key factors in cell–cell fusion during the sexual reproduction of *C. neoformans*.

During sexual reproduction, *C. neoformans* undergoes a cell–cell fusion event before forming hyphae. Our findings suggest that both bilateral and unilateral mating involving the *zfp2*Δ mutants did not produce mating hyphae, indicating that Zfp2 might be important in the cell–cell fusion process that occurs before hyphae development. To further investigate whether Zfp2 is involved in regulating the cell–cell fusion process during the sexual reproduction of *C. neoformans*, we generated the *zfp2*Δ mutants (*MAT*⍺ *zfp2*Δ::*NAT*, *MAT***a**
*zfp2*Δ::*NAT*, Appendix A) with a Nourseothricin Sulfate resistance marker (*NAT*) in two mating-type strains. Successful cell fusion results in growth on a double-selective medium (G418 and Nourseothricin Sulfate), while unsuccessful fusion prevents growth. This approach was used to assess fusion efficiency in *zfp2*Δ mutants during both unilateral (*MAT*⍺ *NEO* × *MAT***a**
*zfp2*Δ::*NAT*, *MAT*α *zfp2*Δ::*NEO* × *MAT***a** *NEO*) and bilateral matings (*MAT*α *zfp2*Δ::*NEO* × *MAT***a**
*zfp2*Δ::*NAT*), compared to wild-type bilateral matings (*MAT*α *NEO* × *MAT***a** *NEO*). The colony count data were normalized so that the wild-type bilateral cell fusion efficiency was 100%. After deleting the *ZFP2* gene, unilateral cell fusion efficiency significantly decreased to only 4–8%. In contrast, bilateral mating resulted in 0% fusion efficiency, demonstrating a highly significant difference compared with the wild-type strains (Figure 3B,C). These results strongly suggest that Zfp2 plays a vital role in regulating the cell–cell fusion process during *C. neoformans* sexual reproduction.

### 3.4. Zfp2 Regulates Key Genes in the Pheromone-Sensing Pathway of C. neoformans

The results show Zfp2 protein is crucial for sexual reproduction in *C. neoformans*, particularly during cell fusion. Subsequently, we measured *ZFP2* expression at different developmental stages with RT-qPCR using mating mixtures of H99 and KN99**a** collected from V8 plates at 0–9 days. Total RNA was purified, and RT-qPCR was performed. Our results showed an overall increase in *ZFP2* expression during early mating, peaking during the initial cell fusion phase (0–24 h) and declining afterward (Figure 1A).

One prerequisite for cell fusion in *C. neoformans*’ mating process is the proper function of the pheromone response pathway, involving Mfα pheromone. Three *Mfα* genes, *MFα1*, *MFα2*, and *MFα3*, were identified in the *MAT*α mating type locus [45], and are expressed when co-cultivated with *MAT***a** cells. The pathway also includes the pheromone receptor and transporter, encoded by *STE3α* and *STE6*, respectively [46,47]. To examine whether Zfp2 affects cell fusion via the pheromone response pathway, we measured the expression of pheromone synthesis (*MFα1*, TL1494/TL1495), sensing (*STE3α*, TL1496/TL1497), and secretion (*STE6*, TL1500/TL1501) genes in Zfp2-related strains using RT-qPCR analysis. The results showed that, even without mating induction, these three genes displayed detectable basal expression in the corresponding *C. neoformans* strains (Figure 4B).

To investigate whether Zfp2 influences the pheromone response in cell fusion, we examined the expression of the three genes (*MFα1*, *STE3α*, and *STE6*) during the sexual reproduction of *C. neoformans*. We found that the pheromones and receptor expression was significantly reduced after 24 h of mating between the *zfp2*Δ mutants and the *ZFP2*^OE^ strains (Figure 4C). This indicates that Zfp2 loss may affect the pheromone response pathway and cell fusion, showing that a fine-tuning of Zfp2 levels is necessary for mating, as both deficiency and excess of the protein impair this process.

To understand the roles of reactive components in *C. neoformans’* pheromone pathway during sexual reproduction, we used the Split marker strategy and developed mutants with the *STE6*, responsible for pheromone transport, labeled *ste6*Δ (TBL462 and TBL463; Appendix A), in the H99 and KN99**a** backgrounds. We also created a *ste3α*Δ mutant (TBL454; Appendix A) in the H99 background, encoding the pheromone response receptor. Unilateral *STE3α* and bilateral *STE6* knockouts failed to form hyphae and basidiospores (Figure 4D). These results, similar to *ZFP2* deletion outcomes, suggest that sexual reproduction halted at hyphal formation, leading us to hypothesize that Zfp2 may regulate cell fusion through the pheromone response pathway.

### 3.5. Zfp2 Plays an Important Role in Fungal Virulence in Mouse and Wax Moth Infection Models

To investigate Zfp2’s role in *C. neoformans’* virulence, we used an inhalation mouse model, inoculating 8-week-old female C57BL/6 mice with the H99 strain, the *zfp2*Δ mutant, *zfp2*Δ::*ZFP2*, and the *ZFP2*^OE^ strain. Mice survival was monitored daily. All mice infected with the wild-type and complemented strains died within 17 to 26 days. In contrast, mice infected with the *zfp2*Δ mutant started dying at 34 days post-infection, with 100% mortality by 64 days. Importantly, mice infected with the *ZFP2*^OE^ strain remained alive for at least 80 days after infection (Figure 5A). 

Since the growth rate of the *zfp2*Δ mutant was significantly slower at 37 °C than that of the wild-type strain, this suggests that the reduced pathogenicity of the *zfp2*Δ mutant during in vivo infection is likely due mainly to its impaired growth at 37 °C. This finding complicates efforts to understand the loss of virulence caused by thermal sensitivity. To explore this further, pathogenicity assays were performed using the wax moth larva model. The results showed a significant decrease in virulence of the *zfp2*Δ mutant at both 30 °C (Figure 5B) and 37 °C (Figure 5C), indicating that the loss of pathogenicity in the *zfp2*Δ mutant is not solely due to decreased thermotolerance.

To investigate the reduced pathogenicity in *zfp2*Δ mutant and *ZFP2*^OE^ strains in mice, we infected C57BL/6 mice intranasally with H99, *zfp2*Δ mutant, and *ZFP2*^OE^ strains. At 7, 14, and 21 dpi, we assessed fungal burden and examined lung, brain, and spleen tissues from five mice per time point. Fungal burden in lungs increased over time with H99, grew more slowly with the *zfp2*Δ mutant, and remained stable with the *ZFP2*^OE^ strain. In brain and spleen tissues, cryptococcal cells were detectable in H99-infected mice from 7 dpi, increased steadily; in *zfp2*Δ-infected mice, yeast cells appeared by 21 dpi; *ZFP2*^OE^-infected mice showed no detectable cells even at 21 days (Figure 6A). 

Histopathological analyses showed that cryptococcal cells appeared in the lung tissue of H99-infected mice by 7 dpi, with tissue damage worsening over time. In contrast, cryptococcal cells were observed only at 14 dpi in *zfp2*Δ-infected mice, and lung tissue damage was milder than in the H99 group. In *ZFP2*^OE^-infected mice, cryptococcal cells were also found at 14 and 21 dpi, but both fungal growth and lung tissue damage were notably lower than in the H99 group.

Histopathological analyses revealed that cryptococcal cells were detectable in the lung tissue of H99-infected mice by 7 dpi, with progressive tissue damage over time. In *zfp2*Δ-infected mice, cryptococcal cells were first observed at 14 dpi, accompanied by a time-dependent increase in inflammatory infiltrates and pneumonic damage that exceeded the wild-type response. Similarly, *ZFP2*^OE^-infected mice exhibited cryptococcal presence at 14 and 21 dpi, though both fungal burden and pneumonic pathology were notably milder compared to the H99 group (Figure 6B). 

Comparative analyses showed that the *zfp2*Δ mutant and the *ZFP2*^OE^ strains both exhibited decreased pathogenicity via distinct mechanisms. The *zfp2*Δ mutant mainly affected the initial lung colonization, limiting fungal growth. In contrast, the *ZFP2*^OE^ strain significantly reduced lung proliferation and lost the ability to invade the brain and spleen. Since cryptococcal cells are eventually cleared in mice infected with the *ZFP2*^OE^ strain, it suggests that Zfp2 influences pathogenicity by regulating cryptococcal colonization, growth, and organ invasion. In summary, both deleting and overexpressing the *ZFP2* reduce *C. neoformans* virulence. *ZFP2* deletion mainly affects early colonization, while overexpression hampers growth and spread, confirming Zfp2 as a key regulatory factor in *C. neoformans* virulence.

### 3.6. Zfp2 Is Involved in the Proliferation of C. neoformans in Macrophage Cells

To clarify how Zfp2-related strains’ pathogenicity varies within the host, we conducted a co-incubation experiment with *C. neoformans* strains and mouse serum at 1 h, 2 h, 3 h, and 4 h to observe how host serum components influence fungal growth (Figure 7A). The results showed no significant survival differences among the *zfp2*Δ mutant, *ZFP2*^OE^ strain, and the wild-type H99, indicating that serum components do not significantly impact their growth.

Recognizing that *C. neoformans* is initially targeted by lung macrophages through phagocytosis, we used a co-culture model with J774 macrophage-like cells to mimic early immune responses in the lung environment. This model aimed to evaluate changes in the growth ability of *C. neoformans* after being engulfed by macrophages. Specifically, sufficient quantities of H99, *zfp2*Δ, *zfp2*Δ::*ZFP2*, and *ZFP2*^OE^ strains were co-incubated with macrophages. After a 2 h incubation, unengulfed *C. neoformans* cells were removed, and the cultures were maintained until designated time points. At 2, 4, and 24 h post-co-culture, intracellular *C. neoformans* cells were released from macrophages by cell lysis, then diluted and plated for colony counting. The results showed no significant differences in phagocytosis levels across the four strains; however, at 24 h, the proliferative ability of the *ZFP2*^OE^ strain within macrophages was significantly lower than that of the other three groups (Figure 7B). To further investigate the in vitro growth characteristics of the strains, their growth was tested in DMEM, revealing no significant differences among *zfp2*Δ, *zfp2*Δ::*ZFP2*, *ZFP2*^OE^, and wild-type H99 (Figure 7C).

In summary, these findings show that after host invasion, the *ZFP2*^OE^ strain has notably decreased growth in macrophages, resulting in poor pulmonary expansion and lower pathogenicity. This explains the regulatory role of the *ZFP2* gene in *C. neoformans’* ability to cause disease.

## 4. Discussion

Zinc finger proteins are significant proteins in eukaryotes, regulating gene expression, protein interactions, and responses to stress [48,49,50,51]. This study focuses on the C_3_HC_4_-type zinc finger protein Zfp2 in *C. neoformans*, which maintains cell membrane and cell wall integrity, as shown by the hypersensitivity of *zfp2*Δ mutants to stress and growth defect at 37 °C. Zfp2 influences virulence factor production; deletion decreases, and overexpression increases capsule size. It is crucial for cell–cell fusion during sexual reproduction, likely by modulating the pheromone response. Additionally, both *ZFP2* deletion and overexpression significantly reduce the virulence of *C. neoformans*.

*C. neoformans*, an opportunistic pathogenic fungus, uses various virulence factors like thermotolerance, capsule synthesis, melanin production, and enzyme secretion (urease and phospholipase) to evade host immunity. Thermotolerance is vital for survival at 37 °C. The polysaccharide capsule resists phagocytosis, blocks antigen presentation, suppresses inflammation, and consumes host complement proteins, weakening immunity [52,53]. This study demonstrates that precise Zfp2 expression is critical for pathogenesis: *zfp2*Δ and *ZFP2*^OE^ strains both exhibit reduced virulence in a mouse model (Figure 5A), despite opposing capsule phenotypes, decreased in *zfp2*Δ and increased in *ZFP2*^OE^ (Figure 2B,C). This dissociation between capsule size and virulence contrasts sharply with Zfp1, a zinc finger protein previously studied by our group, where *ZFP1* overexpression abrogates virulence despite normal capsule/growth [34], underscoring zinc finger proteins’ divergent regulatory strategies.

The contradiction between an enlarged capsule and decreased virulence in the *ZFP2*^OE^ strain is intriguing. This phenomenon, in which increased capsule size in *C. neoformans* paradoxically correlates with attenuated virulence, is well supported by the literature. For instance, studies on *UGE1*/*UGT1* deletion mutants demonstrated capsular enlargement but complete loss of brain colonization and virulence [54], while *SGF29*-deficient strains exhibited pleomorphic capsules yet reduced virulence due to histone acetylation defects [55]. These examples reinforce that capsule size alone does not predict virulence; structural integrity, composition, and regulatory pathways are critical determinants. The *ZFP2*^OE^ strain’s enlarged capsule yet reduced virulence is resolved by macrophage co-incubation assays, revealing impaired intracellular proliferation (Figure 7). This contrasts with Zfp1, where virulence defects persist despite normal capsule/growth, suggesting that Zfp1 regulates immune evasion or stress response pathways beyond capsule synthesis [34]. Both Zfp2 and Zfp1 influence proliferation/dissemination: Zfp2 deletion delays brain/spleen colonization, while *ZFP2* overexpression exhibits severe growth defects in lungs and minimal lesions, ultimately cleared by day 80 (Figure 6). These observations position Zfp2 as a key regulator of fungal fitness within host tissues.

Zfp2 maintains cell wall/membrane integrity, as *zfp2*Δ shows hypersensitivity to Congo red, SDS, and osmotic stress, and cell wall/membrane antifungal drugs caspofungin and fluconazole (Figure 2). Unlike Zfp1, which regulates membrane integrity only (as evidenced by SDS sensitivity) [34], Zfp2 targets both wall and membrane components. These distinct sensitivities suggest Zfp2 and Zfp1 modulate separate cellular compartments, with Zfp2’s role in wall integrity aligning with its broader stress response functions. Other zinc-finger proteins, such as Had1, CnSP1, and Crz1, are also involved in cell wall integrity in *C. neoformans* [56,57,58]. However, these zinc finger proteins differ significantly in sequence, zinc finger domains, and types, making it difficult to deduce how Zfp2 regulates *C. neoformans* cell wall and membrane integrity solely from their functions. Thus, the precise pathways by which Zfp2 influences cell wall and membrane integrity remain unclear and require further research. Future studies might use omics techniques to find genes or proteins, especially those involved in these pathways, that interact with Zfp2.

Zfp2 is indispensable for cell–cell fusion during mating in *C. neoformans*, as evidenced by the complete block of dikaryotic hyphae formation in both bilateral and unilateral matings of *zfp2*Δ mutants. In contrast, *ZFP2* overexpression exhibits context-dependent functionality: it permits fusion in unilateral matings but disrupts hyphal formation in bilateral matings, phenocopying *STE3α*/*STE6* deletion mutants (Figure 3A and Figure 4D). Mechanistically, Zfp2 directly modulates the pheromone response pathway as *zfp2*Δ reduces *MFα1*/*STE3α* expression, while *ZFP2* overexpression suppresses *MFα1*/*STE3α*/*STE6*, distinguishing it from Znf2, which independently promotes pheromone production to drive post-fusion filamentation [33]. In *C. deneoformans,* zinc finger proteins Znf2 and Znf3 similarly regulate cell fusion, with their absence blocking hyphal formation, while Znf2 overexpression specifically enhances filamentation and hyphal morphogenesis after fusion [32,33,59,60,61]. Complementing this, Zfp1 operates at a later stage, governing meiosis: *ZFP1* deletion or overexpression abolishes basidiospore formation and nuclear division [34]. Thus, Zfp2 and Zfp1 act at distinct stages of sexual reproduction, fusion, and meiosis, respectively, with minimal functional overlap. Together, these findings underscore the specialized yet interconnected roles of zinc finger proteins in orchestrating sexual reproduction, with Zfp1, Zfp2, Znf2, and Znf3 acting as pivotal regulators at distinct stages of mating and morphogenesis. Thus, these zinc finger proteins exhibit specialized yet interconnected roles across sexual reproduction stages: Zfp2 and Znf2/Znf3 act early to mediate fusion and morphogenesis, while Zfp1 controls meiotic progression. This staged regulation minimizes functional overlap while ensuring coordinated progression of mating and sporulation. Together, Zfp1, Zfp2, Znf2, and Znf3 exemplify how distinct zinc finger proteins orchestrate sequential steps in sexual reproduction, from initial cell pairing to final spore maturation, highlighting their evolutionary refinement for precise pathogenic and reproductive strategies.

The contrasting phenotypes of Zfp2 and Zfp1 mutants underscore the complex regulatory networks of zinc finger proteins. For Zfp2, reduced virulence in *zfp2*Δ and *ZFP2*^OE^ strains likely stems from disrupted proliferation/dissemination and macrophage survival, with *ZFP2* overexpression exacerbating defects via misregulated downstream targets. Zfp1’s tighter regulatory control, in which both deletion and overexpression impair virulence, suggests its involvement in essential pathogen–host interaction pathways. Future studies should employ CRISPR-based screens and co-immunoprecipitation to map Zfp2’s molecular targets and identify synergistic/compensatory pathways with Zfp1. Integrating these data with existing virulence factor networks will clarify how zinc finger proteins orchestrate pathogenesis. Understanding Zfp2’s role in *C. neoformans* biology has translational relevance. As infections primarily occur via inhalation and progress to fatal meningitis via blood–brain barrier penetration, targeting Zfp2’s regulatory pathways could disrupt virulence while sparing commensal fungi. Given the rising prevalence of antifungal resistance, Zfp2 and related zinc finger proteins represent promising therapeutic targets. Future work should prioritize high-throughput drug screens against Zfp2-associated pathways and validate hits in murine models.

In conclusion, this study highlights Zfp2 as a key regulator of virulence, cellular integrity, and sexual reproduction in *C. neoformans*, a global pathogen that poses a risk to immunocompromised and even healthy hosts due to its evolving pathogenicity. Understanding Zfp2’s unique roles, distinct from those of Zfp1 and other factors, advances knowledge of zinc finger proteins in fungal virulence. With current antifungal treatments limited and resistance growing, exploring Zfp2’s pathways is crucial for new antifungal strategies and preventive measures, addressing a vital health need.

## Figures and Tables

**Figure 1 jof-11-00868-f001:**
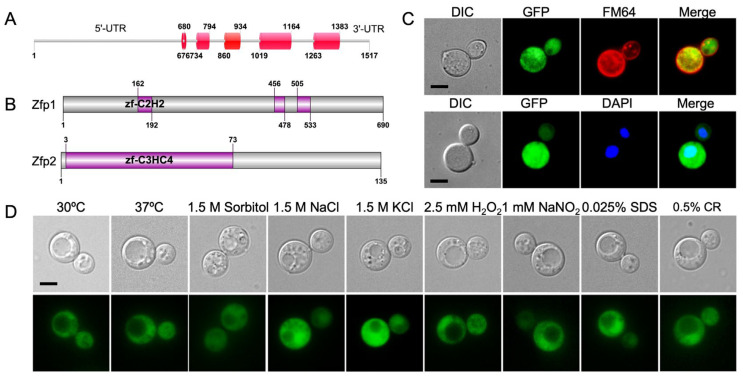
Identification of the zinc finger protein Zfp2 in *C. neoformans*. (**A**) Schematic diagram of the *ZFP2* gene in *C. neoformans*; The *ZFP2* gene has a 678 bp 5′-UTR, 133 bp 3′-UTR, and 5 exons and 4 introns. (**B**) Conserved domains in Zfp1 and Zfp2 proteins of *C. neoformans*. The Zfp1 protein consists of 690 amino acids and contains three C_2_H_2_ zinc finger domains; Zfp2 is composed of 135 amino acids and contains one C_3_HC_4_ zinc finger domain. zf-C_2_H_2_: C_2_H_2_ type zinc finger domain; zf-C_3_HC_4_: C_3_HC_4_ type zinc finger domain; (**C**) Localization of Zfp2 in *C. neoformans*. Yeast cells expressing GFP-Zfp2 were examined using a Zeiss Axio Observer 3 fluorescence microscope. The GFP-Zfp2 fusion protein was observed in the cytoplasm of *C. neoformans* strains. FM4-64: a red fluorescent dye that specifically binds to cell membranes and inner membrane organelles. DAPI: a fluorescent dye that binds to DNA; Bar, 5 μm. (**D**) Localization of GFP-Zfp2 fusion protein in *C. neoformans* under different stress conditions. CR: Congo red. Bar, 5 μm.

**Figure 2 jof-11-00868-f002:**
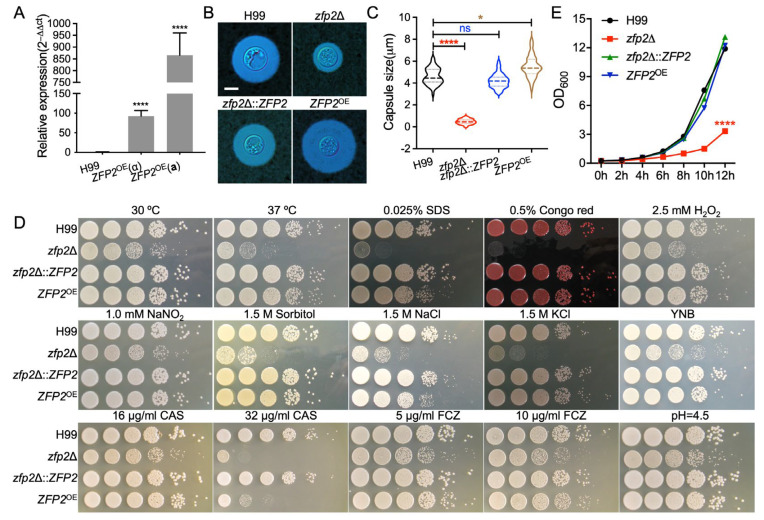
Zfp2’s role in the development of key virulence factors and growth of *C. neoformans*. (**A**) Gene expression levels in the *ZFP2*^OE^ overexpression strains were measured by relative RT-qPCR. ****, *p* < 0.0001. (**B**) Capsule formation was observed after inoculating strains in SAB medium diluted with MOPS, followed by ink staining after induction at 37 °C for 1 day. Bar = 5 μm. (**C**) Capsule sizes of different *Cryptococcus* strains were analyzed using one-way ANOVA. ns, not significant; *, *p* < 0.05; and ****, *p* < 0.0001. (**D**) Growth of *zfp2*Δ mutant and *ZFP2*^OE^ strains under various stress conditions. Overnight cultures of the strains were diluted 10-fold, plated on YPD agar containing various stressors, and incubated at 30 °C for 2–4 days. Conditions are indicated at the top, strains on the left. CAS, caspofungin; FCZ, fluconazole. (**E**) Growth of *zfp2*Δ mutants was tracked in liquid YPD medium at 37 °C with shaking at 200 rpm, and OD_600_ values were recorded every 2 h. Each experiment was performed three times. Error bars show standard deviations. The asterisk indicates the peak growth phase of the *zfp2*Δ mutant relative to the wild-type H99 strain, with a *p*-value of **** (*p* < 0.0001).

**Figure 3 jof-11-00868-f003:**
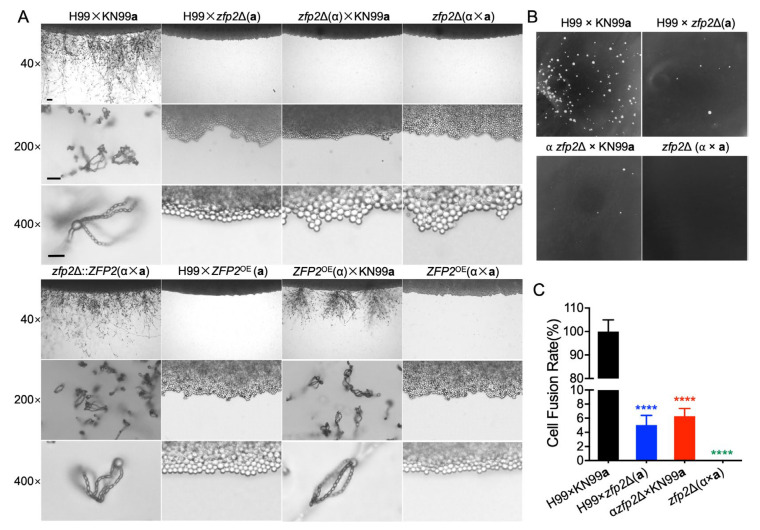
Zfp2’s role in cell–cell fusion during sexual reproduction in *C. neoformans*. (**A**) Unilateral and bilateral mating assays of the wild-type strains (H99, ⍺::*NEO*; KN99**a**, **a**::*NAT*), *zfp2*Δ mutants, and *ZFP2*^OE^ strains were performed on MS media. Mating filament formation and sporulation were imaged at 40× magnification (**top panel**, bar = 100 µm), 200× magnification (**middle panel**, bar = 50 µm), and 400× magnification (**bottom panel**, bar = 20 µm) after 2 weeks of incubation at 25 °C in the dark. (**B**) The cell fusion efficiency of *zfp2*Δ mutants in unilateral and bilateral matings was assessed through quantitative cell–cell fusion assays. H99(YSB119), α::*NEO*; KN99**a**(YSB121), **a**::*NAT*; *zfp2*Δ(α), *MAT*α *zfp2*Δ::*NEO*; *zfp2*Δ(**a**), *MAT***a** *zfp2*Δ::*NAT*. (**C**) Statistical analysis of cell fusion efficiency in *zfp2*Δ mutants was conducted using one-way ANOVA; ****, *p* ≤ 0.0001.

**Figure 4 jof-11-00868-f004:**
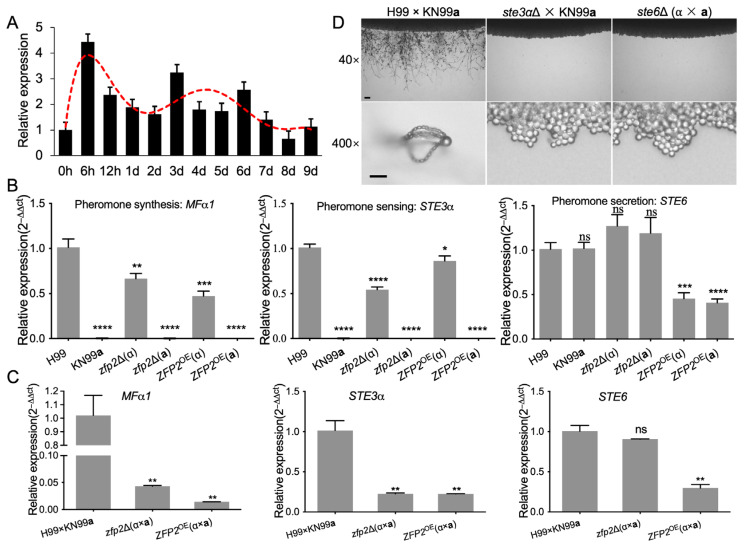
Zfp2 regulates key genes in the pheromone-sensing pathway of *C. neoformans.* (**A**) *ZFP2* expression during mating on V8 medium, measured by RT-qPCR. Mating cultures of H99 × KN99**a** were taken from the plate surface after 0, 6, 12 h, and 1–9 days. RNA was purified, and cDNA was synthesized for RT-qPCR analysis. The comparative CT method was used for relative quantification, with the *GAPDH* gene as the endogenous reference. Error bars represent the standard deviations of three replicates. (**B**) RT-qPCR analysis of gene expression for *MFα1*, *STE3α*, and *STE6* in *C. neoformans* strains. ns, not significant; *, *p* ≤ 0.05; **, *p* ≤ 0.01; ***, *p* ≤ 0.001; ****, *p* ≤ 0.0001. The comparative CT method was used for relative quantification, with *GAPDH* as the endogenous control. (**C**) RT-qPCR analysis of gene expression for *MFα1*, *STE3α*, and *STE6* after 24 h of mating; ns, not significant; **, *p* ≤ 0.01. Relative quantification was performed using the comparative CT method, with *GAPDH* as the internal control. (**D**) Mating hyphae and spore production during the mating of *ste3α*Δ and *ste6*Δ mutants on MS media. The formation of mating filaments and sporulation was observed at 40× (**top panel**, scale bar = 100 µm) and 400× (**bottom panel**, scale bar = 20 µm) magnification after 14 days of incubation at 25 °C in the dark.

**Figure 5 jof-11-00868-f005:**
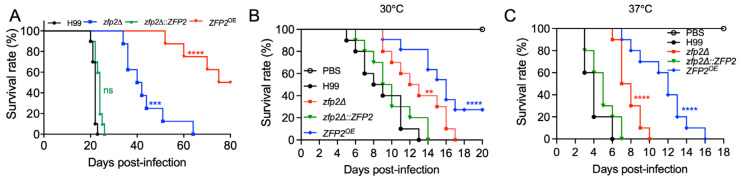
Zfp2 regulates fungal virulence in *C. neofromans*. (**A**) The survival curve of mice infected with *zfp2*Δ mutants. Female C57BL/6 mice (*n* = 10 per group) were intranasally infected with 10^5^ cells of H99, *zfp2*Δ mutant, *zfp2*Δ::*ZFP2* strain, and *ZFP2*^OE^ strain. ns, not significant; ***, *p* ≤ 0.001; ****, *p* ≤ 0.0001 (determined by log-rank [Mantel–Cox] test). (**B**,**C**) Survival curve of *Galleria mellonella* infected with H99, *zfp2*Δ mutant, *zfp2*Δ::*ZFP2* complemented, and *ZFP2*^OE^ strains (*n* = 10 per group) at 30 °C (**B**) or 37 °C (**C**). **, *p* < 0.01; ****, *p* < 0.0001 (determined by log-rank [Mantel–Cox] test).

**Figure 6 jof-11-00868-f006:**
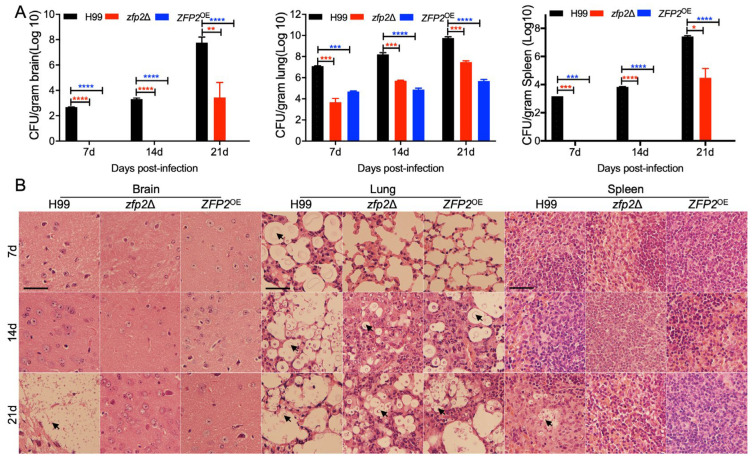
Infection progression in *the zfp2*Δ mutant and the *ZFP2*^OE^ strain in vivo. (**A**) Fungal burden (CFU) was measured in the lung, brain, and spleen. The data show the average ± SD from three animals at each time point; *, *p* ≤ 0.05; **, *p* ≤ 0.01; ***, *p* ≤ 0.001; ****, *p* ≤ 0.0001 (determined by Mann–Whitney test). (**B**) Images of H&E-stained sections of brains, lungs, and spleens infected with *C. neoformans*; arrows indicate cryptococcal cells. Bar = 20 μm.

**Figure 7 jof-11-00868-f007:**
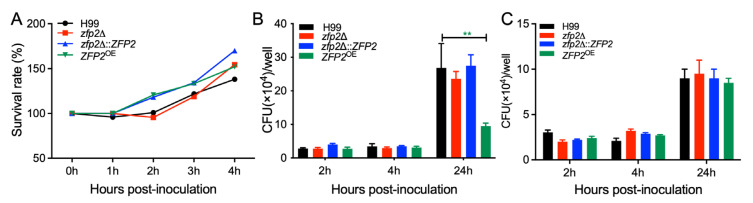
Intracellular proliferation of Zfp2-related strains in macrophage cells. (**A**) The survival rate of each strain after co-incubation with mouse serum is determined by the number of CFU recovered from the culture after a specified period. (**B**) The proliferation of *C. neoformans* within macrophages was assessed using J774 macrophages. After removing non-adherent extracellular yeast cells and incubating for a set duration, the number of CFU recovered from the macrophage culture indicates intracellular proliferation and macrophage killing. **, *p* ≤ 0.01 (determined by Mann–Whitney test). (**C**) The survival rate of each strain after co-incubation with DMEM is measured by CFU count recovered at the specified time.

**Table 1 jof-11-00868-t001:** Partial high-abundance proteins identified in the *atg8*Δ mutants of *C. neoformans*.

Accession	Description	Average *atg8*∆/H99
CNAG_06324	Zinc finger protein Zfp2	2.070858
CNAG_02541	Cyclin-dependent protein kinase inhibitor	1.476625
CNAG_04668	Ubiquitin-conjugating enzyme E2M (Ubc12)	1.471343
CNAG_03759	Conidiation-specific protein 6	1.453521
CNAG_02257	GTP-binding nuclear protein	1.424417
CNAG_06899	26S proteasome regulatory subunit N7	1.379142
CNAG_01362	Cell cycle control protein Cwf19	1.368171
CNAG_02148	Ubiquitin-conjugating enzyme E2 35, variant	1.301707
CNAG_05747	Vacuolar protein sorting-associated protein Vta1	1.300065
CNAG_00508	Vacuolar protein sorting-associated protein Vps17	1.234959
CNAG_05645	COP9 signalosome complex subunit 3	1.215435
CNAG_02167	Vacuolar protein sorting-associated protein Vps27	1.228184

## Data Availability

The datasets presented in this study can be found in online repositories. The names of the repository/repositories and accession number(s) can be found in the article/Appendix A.

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
