# Peer review of "The Zinc Finger Protein Zfp2 Regulates Cell–Cell Fusion and Virulence in *Cryptococcus neoformans"

_jof, 2025, doi:10.3390/jof11120868_

Round 1
Reviewer 1 Report
In their manuscript entitled, “The zinc finger protein Zfp2 regulates cell-cell fusion and virulence in Cryptococcus neoformans”, Fan and colleagues describe the role of the transcription factor ZFP2 in mating and virulence in the critical priority pathogen C. neoformans. The authors characterize ZFP2 thoroughly, including its regulation in response to mating and the aspects of mating that it regulates in turn. Furthermore, they characterize the impact of ZFP2 deletion and overexpression on virulence factor production, cell membrane and cell wall stress responses, and temperature sensitivity. Overall, the article is well written and provides context for this gene’s role in mating in relation to another transcription factor they have previously described, ZFP1, as well as other genes previously shown to directly facilitate mating in this yeast. However, there are two major issues that need addressed, along with some minor grammatical and stylistic issues, prior to publication. These items are listed below.
Major Comments
- In Figure 4C, the data shows that the mfα1∆ mutant still produces basidiospores in a cross with the MATa Furthermore, in Figure S2B, Southern hybridization data indicates the 3.4Kb band corresponding to MFα1 is still present in the H99 “deletion” strains, while a lower ~3Kb band that is also shown in H99 is missing. This is extremely concerning. Prior work by the Heitman lab showed that deletion of MFα1 in C. deneoformans (C. neoformans serotype D in the old taxonomy) resulted in no mating filaments or basidiospores even with a second copy (i.e. MFα2) of the pheromone present (Davidson et al., 2000, Mol Microbiol).
Clearly, both cannot be true without explanation. Either there is a major difference in the activities of the hormone paralogs between these species or there is an issue with the data presented here. The Southern data presented in Figure S2 indicates that the mfα1∆ “deletion” strains in the H99 background are not true deletions since the WT bands are still present. Therefore, I have several questions: 1) how specific is the mfα1∆ deletion construct since there are 3 paralogous pheromone genes in the genome? 2) Does the thicker band located ~3Kb in the H99 lane correspond to the expected size(s) of the other paralogs in the genome? 3) Did you also strip and reblot with NEO-specific probe to see the replacement of the MFα1 gene or document the replacement by PCR (not merely identifying the NEO gene in the transformed strains)? Why do you present Southern data for MFα1 deletion in the KN99a strain when the MATa strain does not possess the MATα-specific genes (e.g. MFα1) and therefore you should not be able to delete this gene in the KN99a strain?
- In Figure 6B, the zpf2∆ and ZPF2OE strains appear to show *increased* inflammation in response to infection compared to WT. Did you measure any of the inflammatory cytokines to quantitatively demonstrate that no increase in inflammation was observed?
Minor Comments
- Line 462: Change “MF1α” to “MFα1”.
- Line 481: Change “MF1α” to “MFα1”.
- Line 483: Change “mf1α∆” to “mfα1∆”.
- Line 507: In Figure 5B, what does the “D” and “S” mean in the figure labels for the overexpression strains. This needs to be indicated in the Figure Legend and in the Methods.
- Line 545: “impairs its” should be roman case, not italics.
- Line 679: Delete “sex” following “union of”. These are just yeast cells, not specialized reproductive cells.
- Line 685: Delete “sex” following “union of”.
- Line 765: Please italicize “C. neoformans”.
- Line 769: Please italicize “C. neoformans”.
- Supplemental Figure 1B & C: Please correct the size indicated by the lower band marker to be “2.1Kb”.
- Supplemental Figure 2B: Please indicate the size of the band immediately below the 3.4Kb band in the figure.

Reviewer 2 Report
The work by Cheng-Li Fan et al describes function of the C. neoformans protein Zfp2.
This protein is a zinc finger protein. Many of these proteins are transcription factors in other fungi but it is unclear what zfp2 is. The paper delivers novel information on this protein, which has not been described to date. The authors conclude that the protein is critical for virulence because it affects the cell integrity under stress, growth at 37 oC and the expression of capsule. The conclusions are based largely on comparing knockout strains. And reconstituted strains. The observation that the overexpression knockout strain is less virulent requires more experiments.
Major points
The results of the overexpression strains are not properly explained and require better controls. There justifications in the discussion are speculations without any data. One control is to generate a second overexpression strain with the same system and the second control should be to generate an overexpression strain under an inducible promoter. Clarify in line 364 that overexpression is not inducible.
It is not clear why the overexpression mutant is not more virulent or at least as virulent as wildtype. It does not have a growth or capsule defect. It seems to be phagocytosed by macrophage. Could there be some erroneous integration of the overexpression plasmid? Virulence unclear if the reduced virulence of zfp2D is only because of slower growth at 37oC. how does the virulence look in galleria at room temperature.
Are zfp1 and zfp2 co-regulated ? nothing is clear or shown about their relationship.
Zfp1 and zfp2 sequence they should be compared are they similar ?
Also the introduction line 64 talks about zinc finger proteins being transcription factors in fungi.
Are there any similarities with other fungal transcription factors. Why did the authors think zfp2 could be a transcription factor? Doesn’t fig 1d not show the protein is expressed on membranes? Based on their presented data I doubt zfp1 and zfp2 are transcription factors and if is it clear what genes they regulate? Can it be predicted in silico? The introduction and discussion don’t really fit.
why is this gene called zfp2-is this justified?
Growth of mutants at lower pH should be checked. Cryptococcus has to survive in the phagolysosome
The intracellular fate of the CN and the zfp2 mutants needs to be better examined especially since the overexpression mutant does not grow why?
In line 564 the state pulmonary macrophages J774 are not pulmonary macrophages they are a cell line
Sensitivity to cell wall antifungal reagents echinocandins should be tested
If zfp2 is located in membranes then azole sensitivity should be checked
Fig 5 it does not make sense to compare CFU and histology at time of death they are expected to be different since the mice dies at different times.
Fig 6 comparing CFU and histology at set times makes more sense
The mating deficit is not really worked out in detail at all.
The discussion needs to be less speculative and more concise
Minor Points
The paper is too long and needs editing. It could be shortened by at least 30% by using a more concise writing style. Some expressions are unusual slide dates imaged not observed e.g.
Many of the references in the introduction e.g. are in correct citations e.g. ref 4 and 5 line 42
see above
Reviewer 3 Report
The present article characterizes a zinc finger transcription factor that regulates an array of downstream targets, playing a role in cell wall and membrane integrity, mating, and virulence. The authors performed an impressive amount of work and generated a very interesting and cohesive dataset. The work is scientifically sound, and the experimental design is thorough. A few points, however, need to be addressed.
The paper mentions Zfp1, but Zfp1/Zfp2 relationship to other zinc fingers is not defined or explored.
The article draws its conclusions mostly from the data obtained from the deletion mutant, failing to explore the apparent dose-dependent effect of Zfp2, as observed in the data obtained from the overexpression mutant. Do the authors believe that overexpression causes disregulation of the pathway? This is again seen in the dichotomy of the enlarged capsule, but reduced virulence observed during overexpression. I'd suggest that the overexpression results need to be better contextualized and discussed. Are there other examples in the literature of bigger capsules leading to decreased virulence?
- Both in the abstract and in the introduction (Lines 38-39), the authors mention that Cryptococcus neoformans infects both immunocompromised and immunocompetent individuals, when C. neoformans actually mainly infects immunocompromised and Cryptococcus gattii can infect immunocompetent individuals. I'd suggest this sentence be rewritten to avoid confusion.
- Line 37: What do the authors mean by yeast-like? C. neoformans is a yeast.
- Line 106-108: I suggest adding the conditions and concentrations tested for the spot assays in the methods section, too. Or are these stressors the ones mentioned in the "2.4 FM4-64 and DAPI Staining" subsection?
- Was sequencing performed to confirm that the sequences of mutant strains (GFP-tagged, complemented, overexpression, and deletion strains) were properly integrated and correct?
- Line 313-316: This explanation of how the target was chosen could also be added to the introduction (in the last paragraph), to help contextualize the study. I'd also suggest contextualizing atg8 and its importance, as it's unclear why this proteomic analysis was chosen, and adding the reference in which this data set was published.
- Figure 1A/1B: Contextualize in the figure legend what each part represents within the sequence. And I believe 1A is in base pairs, while 1B is in amino acids, which should be labeled for easier interpretation.
- Line 327-332/Figure 1C: This result seems out of place here. The remaining results that address mating are in sections 3.3 and 3.4 (Figures 3 and 4). I suggest moving this result.
- Figure 1D: I suggest having the three stains performed on the same cell and merged together, instead of two rows with two stains in each.
- Figure S1 is never called in the text. And the labeling of the kb on Figure S1B is confusing. It is labeled as 3.5 kb twice, and the WT band (supposed to be 3.5 kb) is lower than the supposed 2.1 kb band.
- It would be good to see similar data to Figures S1 and S2 for the complementation and overexpression strains, proving their proper integration.
- Interesting to note that ZFP2 overexpression doesn't show any advantage for growth, even showing some deficiency in SDS and NaCl.
- I'd suggest avoiding the use of the names given to the strains (YSB119, TBL384) in the text. It's confusing for the reader and requires you to go back in the text to see which one is which.
- Line 439-445: This paragraph should have references listed.
- Line 450-451: What does the author mean by "expressed normally", considering there are significant differences between WT, zfp2Δ, and the overexpression strain?
- Line 456-4458: The result doesn't only indicate that "loss of Zfp2 may affect the pheromone response pathway", as overexpression of Zfp2 yielded ever lower expression levels. It demonstrates that a fine-tuning of Zfp2 levels is necessary for mating, as an excess of protein also hinders it.
- Figure S2B: What are the bands shown between 3.4 kb and 2.1 kb? An inespecific band also shows in S2D, which is not explained.
- Figure 4C: Why is ste3 the only one mated against KN99a and not a ste3 mutant in the a mating type?
- I believe that Figures 5B and 4C are unnecessary when you have Figure 6, which is a more scientifically sound method of analyzing the fungal burden.
- A double Zfp1/Zfp2 mutant would be interesting for future studies
- A couple C. neoformans in the discussion are not italicized
Reviewer 4 Report
The manuscript entitled "The zinc finger protein Zfp2 regulates cell-cell fusion and virulence in Cryptococcus neoformans" is well written and provides valuable insights into the role of Zfp2, including a meaningful comparison with Zfp1. The experiments are clearly described, and the results are thoughtfully discussed. The authors have effectively highlighted the study’s limitations and discussed any contradiction in the results section. Overall, the paper showed excellent organization and logical flow.
A few minor comments to enhance the final version of the manuscript:
- Line 47: Replace “⍺ and a” with “MAT⍺ and MATa”
- Please clarify whether this is the first report of Zfp2 deletion and functional investigation in fungi or just C. neoformans.
- Lines 765 and 769: C. neoformans should be italicized.
Round 2
Reviewer 1 Report
Fan and colleagues have address many of the concerns with their manuscript entitled, “The zinc finger protein Zfp2 regulates cell-cell fusion and virulence in Cryptococcus neoformans”. However, some issues yet remain and need to be addressed prior to publication of the manuscript. These items are listed below.
Major Comments
- Section 3.4, Figure 4D: The image showing sporulation in the mfα1∆ mutant is still problematic. The explanation offered by the authors is plausible, but without verification it begs the question as to the rigor of their genetic analysis. Therefore, the references to MFα1 should be excised from the manuscript. The other pheromone pathway genes analyzed in this section support the argument that the pheromone pathway is affected without needing additional support of the MFα1 This would also negate the need to cite and discuss the Davidson et al., 2000, Mol Microbiol paper and its conflicting results. Furthermore, removing the MFα1 data from this manuscript only strengthens the forthcoming manuscript that the authors indicate in their rebuttal on the roles of the three pheromone paralogs in C. neoformans mating.
- 5, Figure 6B: The description of the histological data in the results still doesn’t match the photos in Figure 6B. The zpf2∆ and ZPF2OE strains show *increased* inflammation over time in response to infection compared to WT, which would be described as increased pneumonia (i.e. damage). It is interesting that, while delayed, infection with zfp2∆ leads to increased inflammation while exhibiting “less” capsule production. Perhaps more capsule is being shed as is the case with the rim101∆ mutant (O’Meara, et al., 2010, PLoS Path). This can be addressed with a capsule blot and would provide needed insight into the regulation of virulence factors by Zfp2.
Furthermore, the descriptor “significant” carries analytical weight suggesting statistical analysis of some type was performed. If no such analysis was done, then I suggest changing that adjective to one less statistically charged. The text in lines 504-509 needs to be amended to reflect the data shown in Figure 6B.
Minor Comments
- Line 462: Change “MF1α” to “MFα1”.
- Line 610: Please italicize “ C. neoformans”.
- Line 618: Please italicize “ C. neoformans”.

Reviewer 3 Report
The authors have clarified and improved the manuscript in several significant ways. The connection between Zfp2 and the pheromone pathway could also be strengthened by a few different assays in the future, such as MAPK activation or a reporter assay for the pheromone pathway activity.
- I'd suggest adding scale bars to all the microscopy figures
- In Figure 4, it is stated twice that * < 0.1, when * should be < 0.05. It is important to keep the statistical analysis consistent. If the difference is between 0.05 and 0.1, then there is no statistical difference.
- On Figure 4A, I suggest flipping the 0h, 1h, etc, so they are not on their sides
- In Figure 5, you write the number of larvae in the Galleria groups, but not the number of mice for Figure 4A
- In Figure 6, you again state that * < 0.1, which is incorrect
